# Dynamic Spatio-Temporal Bag of Expressions (D-STBoE) Model for Human Action Recognition

**DOI:** 10.3390/s19122790

**Published:** 2019-06-21

**Authors:** Saima Nazir, Muhammad Haroon Yousaf, Jean-Christophe Nebel, Sergio A. Velastin

**Affiliations:** 1Department of Software Engineering, Fatima Jinnah Women University, Rawalpindi 46000, Pakistan; drsaima.nazir@fjwu.edu.pk; 2Software Engineering Department, University of Engineering and Technology, Taxila 47050, Pakistan; 3School of Electronic Engineering and Computer Science, Queen Mary University of London, London E1 4NS, UK; 4Cortexica Vision Systems Ltd., London SE1 9LQ, UK; 5Computer Engineering Department, University of Engineering and Technology, Taxila 47050, Pakistan; haroon.yousaf@uettaxila.edu.pk; 6School of Computer Science and Mathematics, Kingston University, London KT1 2EE, UK; j.nebel@kingston.ac.uk; 7Department of Computer Science, Universidad Carlos III de Madrid, Leganés, 28911 Madrid, Spain

**Keywords:** human action recognition, Bag of Words (BoW), Bag of Expressions (BoE), spatio-temporal, dynamic neighborhood

## Abstract

Human action recognition (HAR) has emerged as a core research domain for video understanding and analysis, thus attracting many researchers. Although significant results have been achieved in simple scenarios, HAR is still a challenging task due to issues associated with view independence, occlusion and inter-class variation observed in realistic scenarios. In previous research efforts, the classical bag of visual words approach along with its variations has been widely used. In this paper, we propose a Dynamic Spatio-Temporal Bag of Expressions (D-STBoE) model for human action recognition without compromising the strengths of the classical bag of visual words approach. Expressions are formed based on the density of a spatio-temporal cube of a visual word. To handle inter-class variation, we use class-specific visual word representation for visual expression generation. In contrast to the Bag of Expressions (BoE) model, the formation of visual expressions is based on the density of spatio-temporal cubes built around each visual word, as constructing neighborhoods with a fixed number of neighbors could include non-relevant information making a visual expression less discriminative in scenarios with occlusion and changing viewpoints. Thus, the proposed approach makes the model more robust to occlusion and changing viewpoint challenges present in realistic scenarios. Furthermore, we train a multi-class Support Vector Machine (SVM) for classifying bag of expressions into action classes. Comprehensive experiments on four publicly available datasets: KTH, UCF Sports, UCF11 and UCF50 show that the proposed model outperforms existing state-of-the-art human action recognition methods in term of accuracy to 99.21%, 98.60%, 96.94 and 94.10%, respectively.

## 1. Introduction

During the last decade, human action recognition (HAR) has become a well-developing research area and has received substantial attention. It has been applied in a wide range of applications such as sports video analysis, content-based video retrieval, video surveillance, scene recognition, video annotation and human-computer interaction [1]. In particular, there is strong interest in the use of video and depth sensors to monitor the activity of elderly or disabled people in domestic environments to support their independence in a safe environment. The use of dedicated intelligent sensors in such cases could also contribute to maintaining privacy by avoiding the need to stream potentially intrusive video feeds to external human carers. The main aim of action recognition is to classify the action present in a test video irrespective of actors’ locations. Action recognition in realistic scenarios is clearly much more challenging than in constrained environments because background objects can interfere and be mistaken as features of interest.

State-of-the-art approaches have shown notable increases in performance in realistic and complex scenarios. However, there are still some unsolved issues such as background clutter, viewpoint variation, illumination change and class variability [2]. Recently, significant progress has been demonstrated with spatio-temporal feature representation along with variations of the most popular and widely used bag of visual words approaches (BoVW) [3], which have the ability to handle viewpoint independence, occlusion and scale invariance [4,5]. Therefore, there has been a growing interest in exploring the potential of possible variants of the classical BoVW approach, which characterizes actions using a histogram of feature occurrence after clustering [6].

However, since BoVW approaches only consider the frequency of each visual word and ignore the spatio-temporal relation among visual words, such representation is not able to exploit the contextual relationship between visual words. To overcome this drawback, we propose an extension of our previously proposed Bag of Expressions (BoE) model [7], which incorporates contextual relationships of visual words while preserving inherent qualities of the classical BoVW approach. The main idea of visual expression formation is to store the spatio-temporal contextual information that is usually lost during formation of visual words by encoding neighboring spatio-temporal interest points’ information.

Both Kovashka et al. [8] and Gilbert et al. [9] have shown that performance can be improved by the inclusion of neighborhood amongst visual words. Although Reference [8] managed to preserve scale invariance, their solution led to a decline in the capability of handling different viewpoints and occlusions. Alternatively, the inclusion of neighborhoods by Gilbert et al. [9] affected the ability to handle scale and they had to integrate a hierarchy to preserve some scale invariance.

Constructing neighborhoods with a specific number of neighbors, as in Reference [7], can reduce performance in scenarios with occlusion and changing background. For example, under partial occlusion, neighborhoods would contain irrelevant neighbors which would add noise, making them less discriminative. We propose to overcome this by dynamically estimating the number of suitable neighbors, as described in Section 3.4.1, according to the local density of neighboring Spatio-Temporal Interest Points (STIPs) in a Spatio-Temporal Cube (ST-Cube). Figure 1 illustrates the difference between a neighborhood-based approach that uses a fixed number of neighbors with one that uses a spatio-temporal cube’s density. In case of an viewpoint variation, the likelihood of obtaining the same visual expression representation is higher with an ST-Cube based neighborhood.

Kovashka et al. [8] were unable to address viewpoint changes because of restrictions they placed in the formation of neighborhoods. To overcome this, we propose to form a visual expression as an independent pair of visual words and neighboring STIPs. Using the density of a spatio-temporal cube of size *n*, each neighboring STIP in the ST-cube is paired with a visual word to obtain *n* different visual expressions. This provides a local representation of visual expressions and enables some tolerance to viewpoint variation and occlusion.

To support view independence, we represent each action as a visual expression and, as in Reference [7], only the frequencies of these visual expressions are stored. They are formed in such a way that they represent the distribution of visual expressions in the spatio-temporal domain. In contrast to Reference [7], visual expressions are constructed to incorporate the spatio-temporal contextual information of the visual words by combining the visual words and the neighboring STIPs present in each spatio-temporal cube. Similarly, these visual expression representations discard all information related to other visual words and only consider the relationship between a visual word and its neighboring STIPs in the ST-Cube. It focuses on the individual contribution of visual expression and is expected to enhance discriminative power on occlusion. As will be shown here, the proposed model outperforms state-of-the-art results for KTH, UCF Sports, UCF11 and UCF50 datasets.

The rest of the paper is organized as follows. We discuss related work in Section 2. Section 3 describes in detail the proposed dynamic spatio-temporal Bag of Expressions (D-STBoE) model for human action recognition. Evaluation of parameters, experimentation results and discussion are provided in Section 4, followed by conclusions and suggestions for future work in Section 5.

## 2. Related Work

Bag of visual words (BoVW) along with its variations has become a popular approach for many visual interpretation tasks, including human action recognition. In BoVW, video representation is obtained using three steps: feature representation, dictionary learning and feature encoding. In such representation, performance is highly dependent on the discriminative power of each step. As work on action recognition using a BoVW approach is too broad to be covered here, we focus on the related approaches for feature representation and neighborhood features.

Feature representation for video, including holistic and local representation approaches, have been reviewed in detail in survey papers such as Reference [1]. Holistic representations use a global representation of human body, structure, shape and movement [10,11,12], while local representations are based on the extraction of local features [13]. Local representation is favored nowadays because holistic ones are too rigid to capture viewpoint variation, illumination changes and occlusions, which is typical of realistic scenarios [14,15]. Local feature representation emerges as an outcome of the spatio-temporal interest point detector proposed by Laptev et al. [16]. Other traditional feature detectors like Harris corner detector [17], Gabor filtering [18] and the scale invariant feature transform (SIFT) detector [19] are also popular interest point detectors. Based on the popular Harris corner detector, 3D Harris spatio-temporal interest point detector was proposed by Laptev et al. [16] to capture significant variations in both space and time, which has made it particularly applicable to action recognition. An example of such interest points is shown in Figure 2.

Local descriptors are then computed for these detected interest points. Local interest point descriptors can rely on 3D cuboid or trajectories. 3D HoG (Histogram of Gradients) [31], inspired by HoG [32], is used to integrate motion information in the spatio-temporal domain. Histogram of optical flow (HoF) [33] encodes optical flow of pixel value motion in spatio-temporal local regions. Another popular 3D interest point descriptor is 3D SIFT [34].

After computing features, these are further transformed into visual words in a BoVW approach. Compared to text that is usually semi-structured, visual appearance is less structured and exhibits large variations when the same underlying visual pattern is present in varying illumination, different viewpoints and occluded environments. A possible solution to resolve the ambiguity of visual pattern representation is to consider its spatial-temporal context [7].

Yuan et al. [35] introduced the concept of visual phrase to overcome the underlying over-representation and under-representation challenges in visual word representation. Visual phrases are used to represent the spatial co-occurrence of visual words. Meng et al. [36] have used spatio-temporal cues for object search in video and find the top *k* trajectories that are the possible candidates containing the object.

For video representation, Zhao et al. [37] used deep spatial net (VGCNet [38]) and temporal net (Two-Stream ConNets [39]). They extracted the proposed frame-diff layer and convolutional layers and pooled them with trajectory temporal pooling and line temporal pooling strategies. These descriptors were further pooled with VLAD (Vector of Locally Aggregated Descriptors) for video representation. Xu et al. [40] combined a trainable VLAD encoding process and the RCNs (randomly connected neurons) architecture and developed a novel SeqVLAD (Sequential VLAD). Murtaza et al. [41] also proposed a novel encoding algorithm, DA-VLAD, for video representation by exploiting the code-word representation of each action.

Rahmani et al. [42] proposed a novel deep fully-connected neural network for the recognition of human actions from novel viewpoints which they called non-linear knowledge transfer model (R-NKTM). By using dense trajectories for feature representation, their proposed model performed better than traditional methods. In addition, they stated that performance could be further improved by combining multiple feature representation approaches like dense trajectories, HOG, HOF and motion boundary histogram.

Wang et al. [43] trained a deep neural network using spatio-temporal features. They used a Convolution Neural Network (CNN) and Long Short Term Memory (LSTM) units for the extraction of spatial and temporal motion information respectively. Local and semantics characteristics were used to distinguish an object from its background and in addition its spatial information was extracted using a CNN. Different spatial and temporal positions in video were learned using a temporal-wise attention model.

Khan et al. [44] improved the performance of human action recognition methods using a hybrid feature extraction model. For action classification, they used a multilayer neural network on selected fused features and achieved good performance on the KTH dataset.

Many researchers have extended the traditional 2D convolutional neural network to integrate temporal information. 3D CNN was proposed by Hara et al. [45] to include temporal information using a 3D kernel to extract information from both the spatial and the temporal domains. Limiting spatial space looks reasonable whereas, for the extraction of motion information, limiting time domain to a few frames seems deficient. Ng et al. [46] proposed that instead of learning temporal information from few frames, temporal pooling can be applied to integrate the temporal domain information in network layers.

Simonyan et al. [39] extended the convolutional neural network by proposing a two-stream architecture. Similar to the way human vision cortex learns appearance and motion information using ventral (object recognition) and dorsal (motion recognition) streams, they used two pathways to learn spatial and temporal information for appearance and motion respectively. The spatial stream was learned on a model pretrained on ImageNet while the temporal stream was trained on an action recognition dataset to address the unavailability of pretrained models on flow information.

Following the success of the two-stream architecture, Feichtenhofer et al. [47] extended the two-stream CNN and proposed a spatio-temporal residual network. Motivated by the success of ResNet in ILSVRC 2015, they pretrained ResNet on ImageNet for action recognition. They embedded the residual connection between appearance and motion streams to allow the hierarchical learning of spatiotemporal features for both streams.

Hara et al. [45] also utilized the ResNet model for human action classification. They made use of 3D convolutional kernels to integrate temporal information in a residual network. They trained and evaluated their proposed 3D ResNet model on relatively large action recognition datasets (Kinetics and Activity Net). Temporal ResNet [48] also integrated temporal information in a spatial residual network by modifying the basic residual function using an additional temporal conv block.

## 3. Dynamic Spatio-Temporal Bag of Expressions (D-STBoE) Model

In this section, we present a dynamic spatio-temporal Bag of Expressions (D-STBoE) model for human action recognition. The proposed model recognizes human actions performed in complex scenarios by following the pipeline shown in Figure 3. This figure is divided into two sections, that is, training and testing processes. For training, first each video is described using a local feature representation, that is, spatio-temporal interest point’s descriptions. We note that not only is the method independent from the type of features used but it can also be used with features extracted from pre-trained deep networks. For simplicity and to allow comparison with competing methods, here we have used well-established features. Second, for each action class, the widely used k-means clustering algorithm is employed to produce visual words from which a spatio-temporal cube’s density based Bag of Expressions is generated. Third, a histogram of visual expressions is constructed for each class by encoding the training videos’ feature representation using the concatenated visual expression dictionary. Finally, those histograms are used to train a supervised classifier. The testing process is similar: the video of interest is classified by first encoding its feature representation using the visual expression dictionary formed during the training process and then by submitting it to the action classifier. These steps are explained in more detail in the following sub-sections.

### 3.1. Spatio-Temporal Interest Points Detection

The feature detector aims to detect the distinctive locations of interest points which are used as a compact representation of the raw action video. Since in a video spatio-temporal interest points are considered to be both compact and salient for feature representation [14], 3D Harris is selected to detect interest points associated with regions with high-intensity variation in both space and time dimensions. More specifically, we take advantage of the modified Harris corner function proposed by Laptev et al. [16]:(1)$$D=\det\mu -mtrace^{3}\left(\mu \right)$$
where *m* is a tunable sensitivity parameter and μ is a 3 × 3 spatio-temporal moment matrix:(2)$$\mu =G\left(:,\sigma^{2},\tau^{2}\right)*\left\{\begin{array}{ccc} {L_{x}}^{2} & L_{x}L_{y} & L_{x}L_{t}\\ L_{x}L_{y} & {L_{y}}^{2} & L_{y}L_{t}\\ L_{x}L_{t} & L_{y}L_{t} & {L_{t}}^{2} \end{array}\right\}$$
with a spatial scale σ and temporal scale τ. Here *L* is the first order spatio-temporal derivative and *G* is a Gaussian weighting function. As illustrated in Figure 4a where 3D plots show detected interest points for the detected STIPs (blue dots), those points are well localized in both space and time domains. For simplicity detected STIPs are shown in the space domain only in Figure 4b. This helps to obtain a rich representation for each video and enables to differentiate between other action classes and noise. For each video *V*, the set of extracted spatio-temporal interest points are given by $p=\{p_{i}|p_{i}\varepsilon {\left(x,y,t\right)}_{i=1}^{n}\}$, where *n* is the total number of interest points detected for a video *V*. Clearly, the value of *n* varies for each video depending on the number of interest points detected by the 3D Harris corner detector.

### 3.2. STIP Description using 3D SIFT

Feature descriptors are used to describe the detected interest points. Since 3-Dimensional Scale Invariant Feature Transform (3D SIFT) [34], an extension of SIFT [49] in the time domain, is able to represent points such that their representation is invariant in terms of scale, rotation, appearance and occlusion [50], we used that descriptor to characterize the points detected in the previous step. Scovanner et al. [34] obtained the 3D SIFT representation by computing the overall orientation of the neighborhood. In 3D, orientation and magnitude are calculated as:(3)$$Mag\left(x,y,t\right)=\sqrt{\left({L_{x}}^{2}+{L_{y}}^{2}+{L_{t}}^{2}\right)}$$
(4)$$\theta \left(x,y,t\right)=tan^{-1}\left(\frac{L_{y}}{L_{x}}\right)$$
(5)$$\phi \left(x,y,t\right)=tan^{-1}\left(\frac{L_{t}}{\sqrt{{L_{x}}^{2}+{L_{y}}^{2}}}\right)$$
ϕ(x,y,t) encodes the angle away from the 2D gradient direction and θ(x,y,t) represents the angle in the 2D gradient. Once this is computed, the next step is to obtain the 3D SIFT descriptor representation by encoding orientations using sub-histograms. For each video *V*, the feature representations are denoted as $FP=\{f_{i},p_{i}|p_{i}\in {\left(x,y,t\right)}_{i=1}^{n}\}$ where $f_{i}$ is the feature description for the spatio-temporal interest point $p_{i}$.

### 3.3. Class Specific Visual Word Dictionary

K-means clustering is a commonly used algorithm to automatically partition a dataset into *k* groups [1,51]. Let $\mathrm{FP}=\{F_{1}P_{1},F_{2}P_{2},F_{3}P_{3},\ldots ,F_{tm}P_{tm}\}$ represent the obtained features where *tm* is the total number of videos for an action class, *FP* is a set of feature representation for the videos of an action class. We apply k-means clustering to obtain class-specific visual word representations, which correspond to the *k* cluster centers. They become the words of the class-specific dictionary, *CDic*.

### 3.4. Bag of Expressions Generation

After the generation of class-specific visual *word* dictionaries, the next step is to generate a concatenated *visual expression* dictionary containing *C* class-specific visual expression dictionaries, where *C* is the total number of unique action classes.

Constructing a neighborhood with visual words with a fixed number of neighbors [7] affects performance in scenarios with occlusions and changing viewpoints since, for example, in the case of partial occlusion the same neighborhood could contain non-relevant (different class) neighbors, making the neighborhood less discriminative. We propose to set the number of suitable neighbors dynamically, as described in Section 3.4.1, according to the density of a spatio-temporal cube in the frame of interest. However, as such strategy may reduce some viewpoint independence, since it is less probable to obtain neighborhoods containing the same words, this is addressed by describing the neighborhood as an independent pair of neighbors. Each element in a pair contains an independent feature representation with respect to the other paired elements. Each visual word is paired with its neighboring interest points to form visual expressions. Such representation discards all information related to other words and only considers the relation between respective visual word and its neighboring STIP in a pair. Hence, this allows focusing on the individual contribution of a visual expression which should enhance its discrimination power when dealing with occlusions.

A bag of visual expressions is generated by incorporating a visual word’s neighborhood information present in the spatio-temporal cube, as described in the following sub-sections. After the construction of a spatio-temporal cube around a visual word and the calculation of the density of the respective ST-Cube, Section 3.4.1, Section 3.4.2 describes the formation of visual expressions by pairing visual words with their respective neighboring STIPs in the ST-Cube. Finally, Section 3.4.3 explains the formation of a visual expression dictionary by using the *C* class-specific visual expression dictionary.

#### 3.4.1. Spatio-Temporal Cube

For each cluster center, *‘Ccenter’* (visual word), in a dictionary *CDic*, a spatio-temporal cube (ST-C) is formed containing the spatio-temporal interest points around the cluster centers. As shown in Figure 5a, the spatio-temporal cube is aligned with the 3D plane axes. This ST-Cube is constructed to define the neighborhood of the respective visual word to dynamically obtain a suitable neighborhood for each visual word, making it more discriminative. This spatio-temporal cube contains neighboring spatio-temporal interest points from the set *P*, where *P* is the set of interest points detected for an action class (as mentioned in Section 3.1). As shown in Figure 5b, the number of neighbors in a cube can vary and is also dependent on the size of the cube, denoted as *CubeSize*.

The density of each visual word (also referred as cluster center) is calculated in the spatio-temporal cube where ST-C density is defined as:(6)$$Density_{ST-C}=\sum_{j}DenseX\left(p_{j},Ccenter\right)$$
where *DenseX* = 1 if $p_{j}$ falls in the spatio-temporal cube of size $CubeSize^{3}$, otherwise *DenseX* = 0. In other words, visual word density is the number of spatio-temporal neighborhood interest points present in the spatio-temporal cube. Figure 5c illustrates the calculation of visual words’ densities.

#### 3.4.2. Visual Expression Formation

Visual expressions are formed to incorporate the contextual spatio-temporal neighboring information of a visual word, by combining the visual words and their respective neighboring spatio-temporal interest points present in the spatio-temporal cube. Let $p_{q}$ be the neighboring spatio-temporal interest point of the visual word $Ccenter_{1}$. The associated visual expression is formed as:(7)$$VExp_{1q}\left(f,p\right)=\left[Ccenter_{1}\left(f_{Ccenter1},p_{Ccenter1}\right),{fp}_{q}\left(f_{q},p_{q}\right)\right]$$

The number of visual expressions formed against each visual word depends on its $Density_{ST-C}$. For example, as shown in Figure 6, if $Ccenter_{1}$ has $Density_{ST-C}=8$, then the number of visual expressions formed for $Ccenter_{1}$ will be eight and are represented as $\left\{VExp_{11}\left(f,p\right),VExp_{12}\left(f,p\right),VExp_{13}\left(f,p\right),...,VExp_{18}\left(f,p\right)\right\}$.

#### 3.4.3. Visual Expression Dictionary

Similarly to the approach used in Reference [7], after obtaining visual expression representations for each visual word, they are concatenated to form a visual expression dictionary *CEDic*. In contrast to Reference [7], the size of *CEDic* depends on the number of visual words, that is, *k*, obtained for each action class, the number of classes and spatio-temporal cube’s densities. As shown in Figure 7, assuming there are five action classes in an action recognition dataset, after applying k-means clustering (as discussed in Section 3.3), there will be five class specific dictionaries with *k* visual words for each *CDic*. Then, the class-specific visual expression dictionaries, *CEDics*, are generated for each action class. Note that the size of each *CEDic* varies as it is dependent on the size of each visual word’s ST-C density. Finally, these five *CEDic* are concatenated to obtain a visual expression dictionary, which is represented as *EDic*.

### 3.5. Histogram of Visual Expressions Encoding

The proposed model uses histogram encoding [52] for assigning the local feature representation to the corresponding visual expression. Given a set of feature representations $fp=f_{i}p_{i}{|}_{i=1}^{n}$ of a video *V*, let $histE_{s}$ be the assignment of each feature representation $f_{s}p_{s}$ to the corresponding visual expression $VExp_{r}$ such that $histE_{s}=argmin{∥f_{s}p_{s}-VExp_{r}∥}^{2}$, see Figure 8. The histogram encoding vector $HistE\epsilon R^{r}$ is such that ${\left[HistE\right]}_{r}=\left|\{s:histE_{s}=r\}\right|$ where *r* is the corresponding *VExp* in *EDic*.

### 3.6. Action Recognition

Since we use a multi-class support vector machine model for action recognition, we need to select both a kernel function and a multi-class model. We evaluated multi-class SVM models using three different kernel functions, that is, Gaussian kernel, linear kernel and polynomial kernel, which are defined, respectively, as:(8)$$GK\left(x,y\right)=exp\left(-∥x-y^{2}∥\right)$$

(9)$$LK\left(x,y\right)=x^{\prime }y$$

(10)$$PK\left(x,y\right)={\left(1+xy\right)}^{2}$$

Multi-class SVMs models use different types of coding design schemes to learn [53]. A video of interest is assigned the action class label $\hat{C}$ that maximizes aggregation of the losses for the *L* binary learner and defined as:(11)$$\hat{C}=\underset{c}{argmin}\sum_{l=1}^{L}\left|M_{cl}\right|g\left(M_{cl},S_{l}\right)$$
where *M* is the coding design scheme matrix with elements $M_{cl}$, $S_{l}$ and is the predicted classification score for the positive class of learner *l* and *C* is the total number of unique action classes. Six different coding design metrics were considered for learning a binary learner:
**One vs. one:** The multiclass model learns *C(C − 1)* binary learners. For this coding design matrix, one class is positive, another is negative and the rest is ignored.**One vs. All:** For each binary learner of a total *C* binary learners, one class is positive and the rest is negative.**Ordinal:** Matrix elements are assigned such that for the first binary learner the first class is positive and others are negative. For the second binary learner, the first two classes are positive and the rest are negative and so on. It has *C − 1* binary learners.**Binary Complete:** The multi-class SVMs model learns $2^{c-1}-1$ binary learners such that it assigns 1 or −1 to each class provided that at least one class is assigned positive and one class is assigned negative.**Dense Random:** Each element of the design matrix is assigned equal probability and has approximately $10log_{2}C$ binary learners.**Sparse Random:** It assigns 1 or −1 to the class with 0.25 probability and 0 to class with 0.5 probability. It has approximately $15log_{2}C$ binary learners.

The dynamic spatio-temporal based Bag of Expressions (D-STBoE) method is summarized in Algorithm 1.

**Algorithm 1** D-STBoE algorithm for Human Action Recognition
**Require:** Training video dataset $VS_{train}=\left\{VS_{1},VS_{2},VS_{3},...,VS_{c}\right\}$ where *C* is the number of action classes,
    Training labels $Label_{train}$,    Testing video dataset $VS_{test}$1:
EDic←[]
2:
**for**
*z = 1:c*
**do**
3:    **for each** video in $VS_{z}$
**do**4:        obtain feature representation *FP*5:    do k-mean clustering to obtain $CDic_{z}$6:    **for each** visual word $Ccenter_{w}$ in $CDic_{z}$
**do**7:        **for**
$u=1:Density_{Ccenter_{w}}$
**do**8:           compute $VExp_{uw}$9:        $CEDic_{z}=concatenate\left[CEDic_{z},VExp_{uw}\right]$10:    $EDic=concatenate[EDic,CEDic_{z}]$11:**for each** video *v* in training set $VS_{train}$
**do**12:    feature encoding using *EDic* to get $HistE_{train}$13:
*TrainC = TrainClassifier($HistE_{train}$,$Label_{train}$)*
14:**for each** video $V_{test}$ in training set $VS_{test}$
**do**15:    obtain feature representation *FP*16:    feature encoding using *EDic* to get $HistE_{test}$17:    $Label_{test}$ = Predict(*TrainC*,$HistE_{test})$   **return**
$Label_{test}$



## 4. Experimentation Results and Discussion

### 4.1. Datasets

The proposed model was evaluated on four benchmark datasets for human action recognition, that is, KTH, UCF Sport, UCF11 and UCF50. Although **KTH** [21] contains actions performed in simple scenarios and as a consequence it is not considered as challenging by state-of-the-art human action recognition methods, it is included for comparison with related work. It consists of videos involving a single actor performing single actions. It contains 6 different types of actions. They are recorded in a constrained environment with a static background. The dataset is divided into training and testing videos based on actor identity [21]. Videos involving 16 different actors are used for training, while the remaining, involving 9 different actors, are used for testing. Performance is reported using average accuracy over all classes.

The **UCF Sports** [25] dataset comprises 150 videos taken from actual sports broadcasts. It contains 10 different types of actions. It is a challenging dataset since it is captured in realistic environments which include wide ranges of views and scenes. Since each action is performed by different actors, each individual action is performed in many different manners. We used the leave-one-sample-out validation scheme: one original video is used for testing and the other 149 videos for training. Performance is evaluated according to average accuracy, that is, accuracy over all classes.

**UCF11** [28], previously known as the YouTube dataset, is a very challenging dataset as it contains videos taken from the web. They are captured in unconstrained environments and include challenges such as large variation of illumination changes, a mix of steady and shaking cameras, cluttered backgrounds, low resolution and variation of object scale. It comprises 11 types of actions. Each action class has 25 groups, each containing at least 4 video clips. Each group’s video clips share features such as having been captured in a similar environment from the same viewpoint and having been performed by the same actor. Similar to Reference [25], we used the leave-one-group-out validation method and performance is reported using average accuracy over all classes.

**UCF50** is also a very challenging dataset captured in realistic environments with large variations in viewpoints, backgrounds, camera motions, object appearances and poses. Similiarly to UCF11, for each action class in UCF50, video clips are grouped into 25 groups each containing at least 4 video clips. Each group shares some features, like similar environment, same actor and similar viewpoints. As proposed in References [25,30], a Leave One Group Out evaluation method is used for evaluation and performance is reported using average accuracy over all classes.

### 4.2. Feature Extraction and Parameter Tuning

We have extracted spatio-temporal interest points using the 3D Harris STIP detector with standard values, that is, σ = 4, τ = 2 and *m* = 0.001 [15]. These points are described by 3D SIFT with 2056 dimensions. We optimized parameters involved in the proposed model on UCF Sports dataset as it is a challenging dataset containing a wide range of actions and scenes. We performed a set of experiments to choose the best *k* visual words representation for the class-specific dictionary.

We also evaluated the proposed D-STBoE model performance by varying ST-Cube size. Figure 9 shows that the increase in class-specific dictionary (CDic) size (i.e., value of *k*) results in improved performance irrespectively of ST-Cube size. It also reveals that an increase in ST-Cube size tends to decrease performance after ST-Cube size = 5. Indeed, a larger size is likely to increase the number of non-relevant neighbors contributing to the visual expression formation, hence making it less discriminant. However, a smaller ST-Cube size may fail to integrate relevant neighbors, hence making the visual expression incomplete. Performance is even more dependent on the *’k’* parameter as an insufficient dictionary size leads to poorer accuracy. As our experiments show that up to *k* = 300 there is significant performance improvement, that value was selected. Similarly, an ST-Cube Size of 5 was chosen to produce optimal performance.

To analyze the computational complexity of the proposed method, we have evaluated the impact of the parameter that has the most significant contribution in terms of computational time. Following measurements of the average running time for the generation of a class-specific dictionary as described in Section 3.3, Table 1 shows that an increased number of cluster’s centers *(k)* for k-means clustering leads to increased processing time.

In the next set of experiments, we evaluated the performance of different multi-class SVM models used for action recognition by testing three different kernel functions and six coding design matrices for the binary learner, as outlined in Section 3.6. As expected from Reference [54], results show that the Gaussian kernel outperforms linear and polynomial kernels, see Table 2. Moreover, the One versus All coding design matrix shows better performance than the other matrices.

### 4.3. Contribution of Each Stage

This section considers the contribution of each stage in the proposed model, as tested on the UCF Sports dataset. As presented in Section 3.3, we used a supervised dictionary learning approach (using k-means clustering) to learn the class-specific visual word representation for each class. Here, we evaluate the impact of learning the *’C’* class specific dictionaries versus one unsupervised dictionary. Figure 10 shows that a significant improvement is brought by using a supervised dictionary learning approach in comparison with the unsupervised one. Although an increase in dictionary size leads in both cases to improved performance, the supervised approach outperforms the unsupervised one consistently.

The trend line in Figure 11 shows that the density of spatio-temporal varies approximately quadratically with ST-Cube size. Therefore, we can conclude that an increase in cube size will also result in an increase in visual expression dictionary size.

However, this trend does not result in increased accuracy, as illustrated by Figure 12. If we compare this trend with the effect of visual word dictionary size on performance (refer to Figure 10), we can conclude that there is no relationship between visual expression dictionary size and visual word dictionary size in terms of achieved performance.

### 4.4. Comparison with the State-of-the-Art

The experiment section concludes with a comparison of the proposed model with state-of-the-art methods, including recent deep learning based methods. It should be noted that D-STBoE parameters were optimized for UCF Sports dataset as discussed in Section 3.4.2.

#### 4.4.1. Evaluation on the KTH Dataset

Table 3 shows that, in terms of average accuracy, D-STBoE and our previously proposed BoE perform better than all state-of-the-art methods with the exception of the Multilayer neural network, which achieved a slightly better performance of 99.80% against 99.21% for D-STBoE.

#### 4.4.2. Evaluation on the UCF Sports Dataset

As shown in Table 4, D-STBoE outperforms all state-of-the-art methods, achieving an additional 2.6% accuracy when compared with the second best performer. This success should, however, be tempered by the fact that D-STBoE was optimised for this dataset. It should also be noted that two other ’traditional’ methods outperform the deep learning approaches on this dataset.

#### 4.4.3. Evaluation on the UCF11 and UCF50 Datasets

We compared the proposed model with the state of the art for the UCF11 and UCF50 datasets, with successful methods that have proven particularly suitable in dealing with the many practical problems represented in these realistic datasets. As shown in Table 5, our method outperforms significantly all the state-of-the-art approaches for UCF11 dataset with the exception of a deep neural network, which delivers an accuracy 1.82% higher.

Table 6 shows the comparison of proposed approach for UCF50 dataset. Even in this very challenging and realistic dataset, the D-STBoE model shows reliable recognition performance. This further indicates that the class-specific visual expression formation is discriminative and that the D-STBoE model is effective for action recognition.

A particular strength of the proposed model is that it shows some degree of tolerance to occlusion and viewpoint variation. Figure 13 and Figure 14 show some positive examples of the ’Basketball’ action (UCF11 dataset) captured from different viewpoints and ’Spiking’ action (UCF11 dataset) in a non-occluded and occluded environment respectively. As the proposed D-STBoE model generates a visual expression based on cuboid neighborhood density, it only focuses on relevant local information which enhances its discrimination power.

## 5. Conclusions and Future Work

In this paper, a dynamic spatio-temporal Bag of Expressions model for human action recognition has been introduced. Not only does it preserve the inherent qualities of the classical BoVW approach but it also acquires contextual information that would be lost with a conventional formation of visual words representation. We proposed a new Bag of Expressions generation method based on the density of spatio-temporal cube of visual words. The new model addresses the inter-class variation challenge observed in realistic scenarios by using a class-specific representation for visual expression generation. Viewpoint variation and occlusion challenges are also partially addressed by considering the spatio-temporal relation of visual words representation. The model was tested on four publicly available datasets that is, KTH, UCF Sports, UCF11 and UCF50. D-STBoE consistently outperformed all state-of-the-art ’traditional’ methods while achieving competitive results when compared to deep learning approaches, which demonstrated best performance on two of the four datasets. In terms of practical deployment, D-STBoE could be a competitive solution compared to deep learning based systems, since, unlike them, it does not require a large dataset for training that is, D-STBoE is better suited for practical applications where availability of training data is usually limited.

Future work will explore the potential advantage of using spatio-temporal cube’s density based neighborhood information for deep learning features and methods and also investigate exploiting spatio-temporal neighborhood information for action detection in even more realistic and complex scenarios.

## Figures and Tables

**Figure 1 sensors-19-02790-f001:**
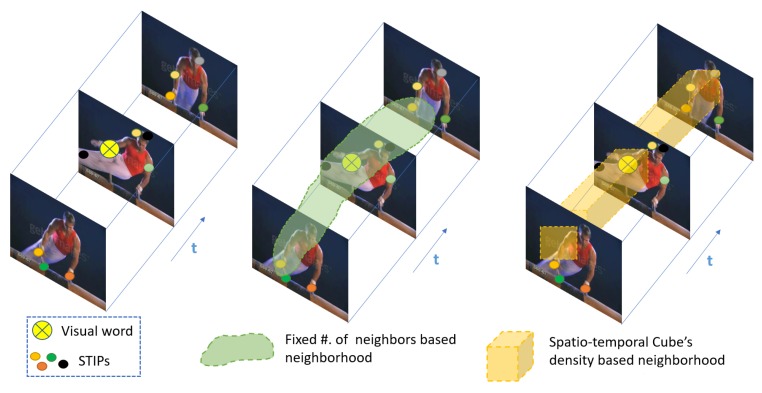
Illustration of STIPs (**left**), neighborhoods based on a fixed number of neighbors (**middle**) and a spatio-temporal cube’s density (**right**)). For a fixed number of neighbors, each visual word neighborhood is formed based on the same number of STIPs falling in that neighborhood. When using spatio-temporal cube’s density, the neighborhood is formed based on the number of STIPs falling in the spatio-temporal cube (ST-Cube).

**Figure 2 sensors-19-02790-f002:**
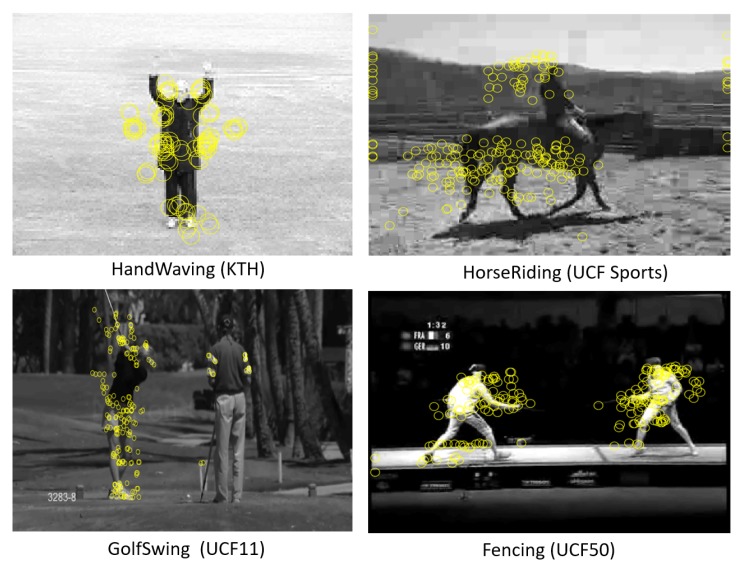
Detected Spatio-temporal interest points (shown in the space domain only, for clarity). Examples from KTH [16,20,21,22,23], UCF-Sports [24,25,26], UCF-11 [27,28] and UCF-50 [29,30] datasets.

**Figure 3 sensors-19-02790-f003:**
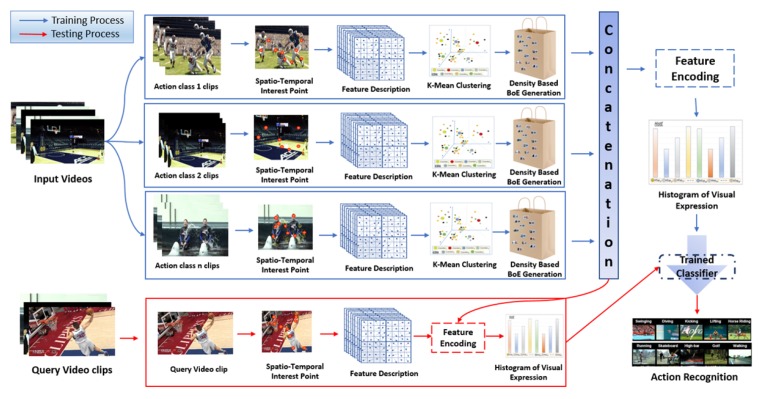
Dynamic Spatio-temporal Bag of Expressions (D-STBoE) Model for human action recognition.

**Figure 4 sensors-19-02790-f004:**
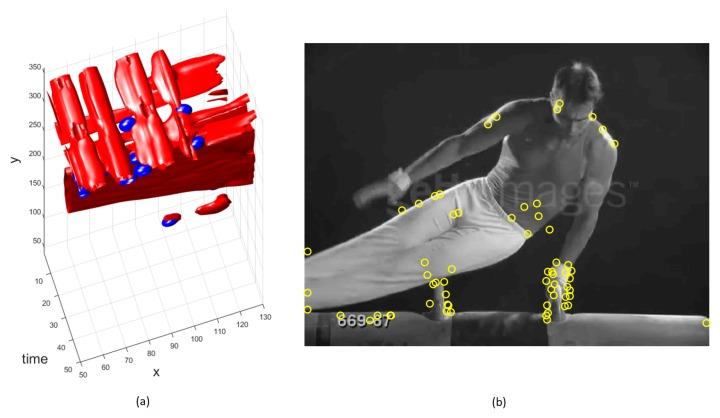
Spatio-temporal interest points for the Swing-Bench action from the UCF Sports dataset [24] (**a**) 3D Plot of detected interest points for Swing-Bench action (**b**) Pictorial representation of detected interest points for Swing-Bench action, shown only in the space domain for simplicity.

**Figure 5 sensors-19-02790-f005:**
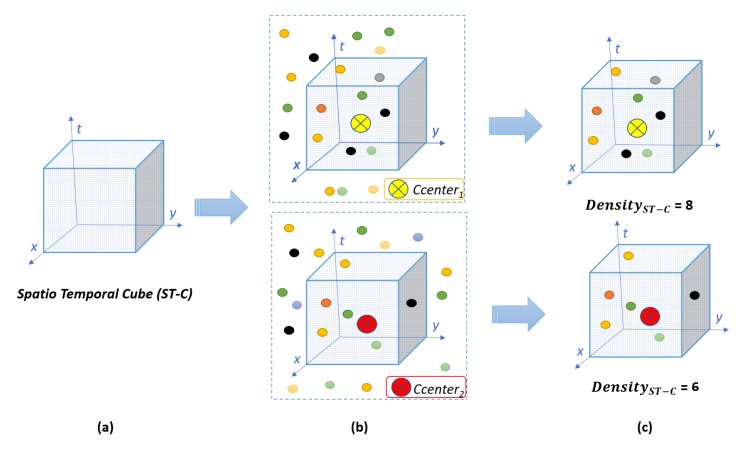
(**a**) Spatio-temporal Cube aligned with 3D plan axes (**b**) Neighboring spatio-temporal interest point around cluster centers (**c**) visual word density calculation.

**Figure 6 sensors-19-02790-f006:**
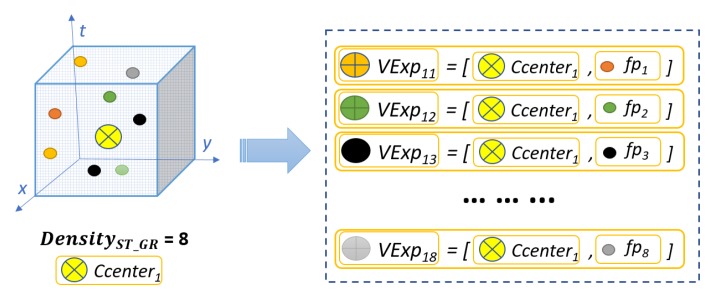
Visual expression formation for a visual word $CCenter_{1}$ of density 8.

**Figure 7 sensors-19-02790-f007:**
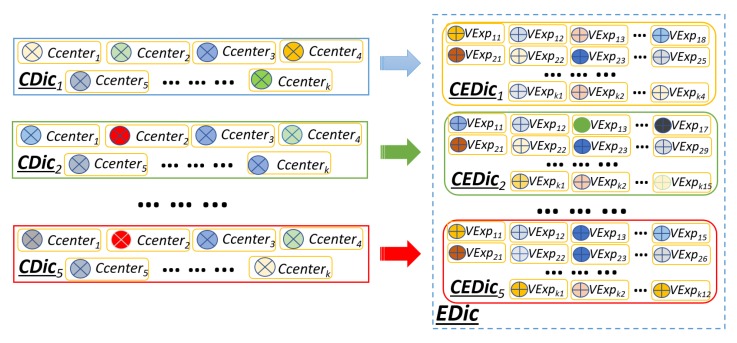
An example of visual expression dictionary *(EDic)* generation from class specific dictionaries *(CDics)*.

**Figure 8 sensors-19-02790-f008:**
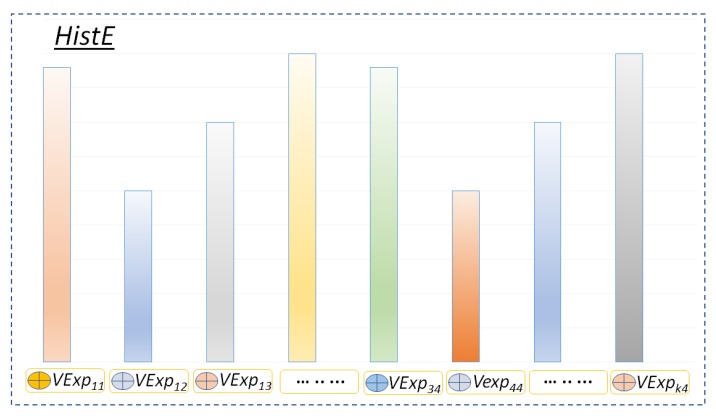
Histogram of visual expression representation.

**Figure 9 sensors-19-02790-f009:**
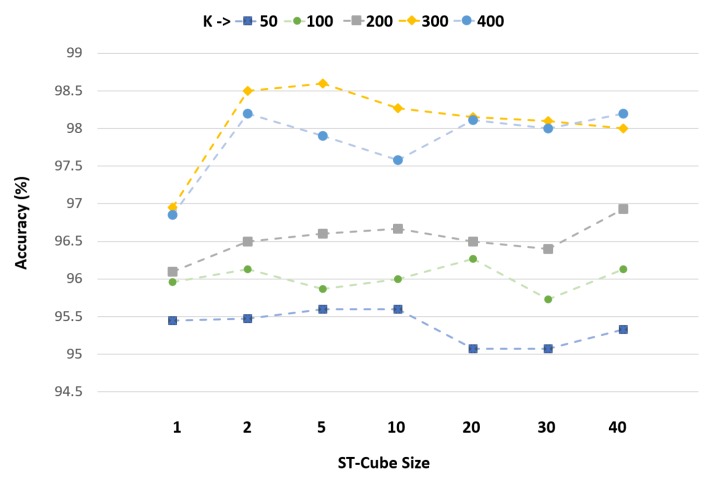
*k* and *ST-Cube size* parameter evaluation for proposed D-STBoE model on the UCF Sports dataset.

**Figure 10 sensors-19-02790-f010:**
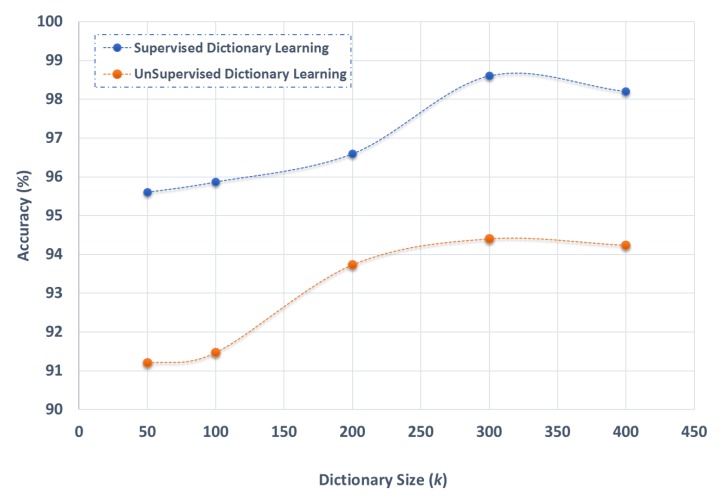
Evaluation of dictionary learning approach on the UCF Sports dataset.

**Figure 11 sensors-19-02790-f011:**
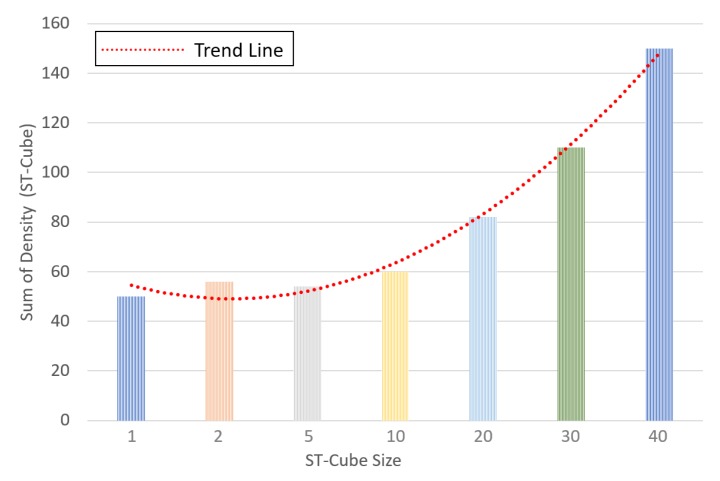
Relationship between ST-Cube size and sum of ST-Cube Density.

**Figure 12 sensors-19-02790-f012:**
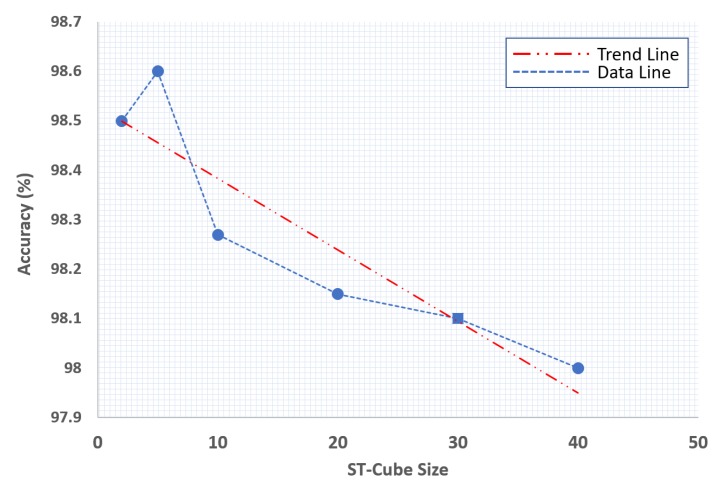
Spatio-temporal cube size evaluation.

**Figure 13 sensors-19-02790-f013:**
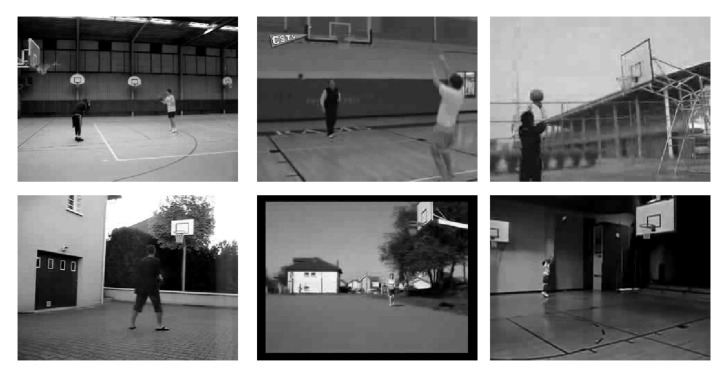
Positive examples of ’Basketball’ action (UCF11 dataset [27]) captured from different viewpoints.

**Figure 14 sensors-19-02790-f014:**
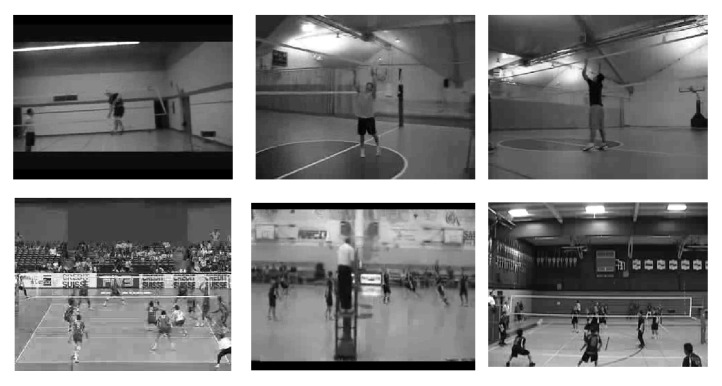
Positive examples of ’Spiking’ action (UCF11 dataset [27]) in non-occluded and occluded environment.

**Table 1 sensors-19-02790-t001:** Computation time with respect to the class specific dictionary size (*k*).

Size (CDic)	Time (s)
K = 50	92.20
K = 100	198.35
K = 200	369.45
K = 300	524.12

**Table 2 sensors-19-02790-t002:** Evaluation of Multi-class SVM model for action recognition on the UCF Sports dataset.

Coding Design Matrix/	Gaussian	Linear	Polynomial
Kernel Function	Kernel	Kernel	Kernel
One vs. One	96.7	95.5	94.5
One vs. All	98.6	97.1	96.7
Ordinal	95.7	94.7	94.0
Binary Complete	97.7	96.4	95.8
Dense Random	98.5	97.2	96.6
Sparse Random	98.5	97.7	96.7

**Table 3 sensors-19-02790-t003:** Accuracy evaluation of our method compared to state-of-the-art methods on the KTH Dataset.

Author	Method	Results
Proposed	Dynamic Spatio-temporal Bag of Expressions (D-STBoE) Model	99.21
[7]	Bag of Expression (BoE)	99.51
[44]	Multilayer neural network	**99.80**
[55]	Foreground Trajectory extraction method	97.50
[56]	Power difference template representation	95.18
[57]	Direction-dependent feature pairs and non-negative low-rank sparse model	94.99
[58]	Space-time robust representation	94.60

**Table 4 sensors-19-02790-t004:** Accuracy evaluation of our method compared to state-of-the-art methods on the UCF Sports Dataset.

Author	Method	Results
Proposed	Dynamic Spatio-temporal Bag of Expressions (D-STBoE) Model	**98.60**
[7]	Bag of Expression (BoE)	97.33
[43]	Spatio-temporal features with deep neural network	91.80
[42]	Robust non-linear knowledge transfer model (R-NKTM)	90.00
[59]	Universal multi-view dictionary	91.00
[55]	Foreground Trajectory extraction method	91.40
[60]	Holistic and motion cues based approach	95.60
[61]	3D-TCCHOGAC and 3D-HOOFGAC feature descriptors	96.00
[56]	Power difference template representation	88.60
[62]	Factored matrices of dual tensors	92.70
[63]	Multi-Region two stream R-CNN	95.50
[64]	Spatio-temporal tube of maximum mutual information	90.70
[65]	Local motion and group sparsity-based approach	89.70
[66]	Dense trajectories and motion boundary descriptors	88.00
[67]	Sparse representation and supervised class specific dictionary learning	86.60
[68]	Invariant spatio-temporal features with independent subspace analysis	86.50
[69]	Person detector along with BOF approach	86.70
[8]	hierarchy of discriminative space-time neighborhood features	87.30

**Table 5 sensors-19-02790-t005:** Accuracy evaluation of our method compared to state-of-the-art methods on the UCF11 Dataset.

Author	Method	Results
Proposed	Dynamic Spatio-temporal Bag of Expressions (D-STBoE) Model	96.94
[43]	Spatio-temporal features with deep neural network	**98.76**
[59]	Universal multi-view dictionary	85.90
[55]	Foreground Trajectory extraction method	91.37
[70]	Graph-based multiple-instance learning	84.60
[65]	Local motion and group sparsity-based approach	86.10
[66]	Dense trajectories and motion boundary descriptors	84.10
[68]	Invariant spatio-temporal features with independent subspace analysis	75.80

**Table 6 sensors-19-02790-t006:** Accuracy evaluation of our method compared to state-of-the-art methods on the UCF50 Dataset.

Author	Method	Results
Proposed	Dynamic Spatio-temporal Bag of Expressions (D-STBoE) Model	**94.10**
[71]	HMG + iDT Descriptor	93.00
[72]	Bag of Words and Fusion Methods	92.30
[5]	Dense Trajectories	91.70
[66]	Dense Trajectories and motion boundary descriptor	91.20

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
