# Peer review of "Dynamic Spatio-Temporal Bag of Expressions (D-STBoE) Model for Human Action Recognition"

_sensors, 2019, doi:10.3390/s19122790_

Round 1
Reviewer 1 Report
The paper investigates the use of Dynamic Spatio-Temporal Bag of Expressions in the field of Human Action Recognition. The approach is an extension of the very well know technique named Bag of Visual Words (BoVW). In a previous work, the author tested a first extension called Bag of Expressions (BoE) by associating neighboring interest points to maintain contextual spatio-temporal relationship between words. In this paper, authors are aiming to solve reduced performance problems due to occlusions and changing background by dynamically estimating the number of neighbors. Using time as 3rd dimension, a modified version of 3D Harris detector has been used to identify Spatio-Temporal Interest Points (STIPs) and represented as 3D SIFT. K-Means algorithm has been used to generate class-specific visual word representations to be used to generate BoE. A Spatio-Temporal Cube (ST-Cube) is created from the STIPs around the each cluster center to dynamically obtain a suitable neighborhood for each visual word. From the spatio-temporal neighboring information, the actions are classified by using a histogram of visual expression representation and a multi-class SVM classifier. Tests have been performed on 4 datasets by comparing the proposed algorithm with state of the art solutions. The performance are significant and very close or better than best existing solutions.
The paper is generally good, the proposed method is interesting and experiments are well design. I would suggest authors to simplify the writing, that sometimes is hard to follow especially in Section 3. In particular, it would help using in this Section an improved set of images better explaining the steps initially described in Figure 3. Moreover, authors should better describe the differences between training and testing processes illustrated, again, in Figure 3. Personally, Figure 4 and Figure 5 are difficult to read. Table 1 is very large. The font dimension should be aligned with the remaining of the paper.
Overall, a very interesting work.
Author Response
Many thanks for your useful review. Please see attached response

Reviewer 2 Report
This paper proposes a spatio-temporal bag of expressions model for human action recognition. The proposed model could be taken as a new hand-crafted video feature learning method for human action recognition. The proposed method is technically sound. The presentation is clear to follow. I have some main concerns:
1.Authors are encouraged to discuss more related spatio-temporal feature learning methods proposed recently, so as to better position the proposed method, e.g.,
-- Pooling the Convolutional Layers in Deep ConvNets for Video Action Recognition. IEEE Trans. Circuits Syst. Video Techn. 28(8): 1839-1849 (2018)
-- Sequential Video VLAD: Training the Aggregation Locally and Temporally. IEEE Trans. Image Processing 27(10): 4933-4944 (2018)
2. Authors are encouraged to evaluate the proposed method on HMDB or ActivityNet, which are more challenging.
3. How about the time complexity and memory efficiency of the proposed method? It would be better to report some empirical results.
Author Response

(The authors gave the same response as above.)

Round 2
Reviewer 2 Report
Authors have addressed most of my concerns. I understand that the revision time is limited for new experimental results on new video dataset, however, it would be better to provide the results in the final version.
Author Response
Dear reviewer,
We appreciate your encouraging comments. As the journal works on a very rapid turn around basis and this type of algorithm requires time to process and verify datasets such as HMDB-51 and ActivityNet, we are afraid that it will not be possible to generate results to report them in the current paper to be on time for the journal's tight deadline (especially when we consider that we are on different countries with different time zones!). We believe that by making our work public now this will allow other researchers to consider it as a baseline for their work. Please also rest assured that we plan to submit results on the suggested datasets for a follow-up paper as soon as we can. We trust that this will be satisfactory to proceed with this paper.